# Neurological manifestations of scrub typhus infection: A systematic review and meta-analysis of clinical features and case fatality

**Ali M. Alam**[1,2]*, **Conor S. Gillespie**[3], **Jack Goodall**[4], **Tina Damodar**[5], **Lance Turtle**[4,6,7], **Ravi Vasanthapuram**[5], **Tom Solomon**[6,7], **Benedict D. Michael**[1,7,8]

1 Department of Clinical Infection Microbiology and Immunology, Institute of Infection, Veterinary, and Ecological Science, University of Liverpool, Liverpool, United Kingdom, 2 Barts Health NHS Trust, London, United Kingdom, 3 Department of Clinical Neurosciences, University of Cambridge, Cambridge, United Kingdom, 4 Tropical & Infectious Disease Unit, Liverpool University Hospitals NHS Foundation Trust, Liverpool, United Kingdom, 5 Department of Neurovirology, National Institute of Mental Health and Neurosciences, Bangalore, India, 6 The Pandemic Institute, Liverpool, United Kingdom, 7 The NIHR Health Protection Research Unit in Emerging and Zoonotic Infections, Liverpool, United Kingdom, 8 Department of Neurology, The Walton Centre NHS Foundation Trust, Liverpool, United Kingdom

* ali.alam@liverpool.ac.uk

**Data Availability Statement:** All data are available in the manuscript and the Supporting Information files.

## Abstract

### Background

Scrub typhus has become a leading cause of central nervous system (CNS) infection in endemic regions. As a treatable condition, prompt recognition is vital. However, few studies have focused on describing the symptomology and outcomes of neurological scrub typhus infection. We conducted a systematic review and meta-analysis to report the clinical features and case fatality ratio (CFR) in patients with CNS scrub typhus infection.

### Methods

A search and analysis plan was published in PROSPERO [ID 328732]. A systematic search of PubMed and Scopus was performed and studies describing patients with CNS manifestations of proven scrub typhus infection were included. The outcomes studied were weighted pooled prevalence (WPP) of clinical features during illness and weighted CFR.

### Results

Nineteen studies with 1,221 (656 adults and 565 paediatric) patients were included. The most common clinical features in CNS scrub typhus were those consistent with non-specific acute encephalitis syndromes (AES), such as fever (WPP 100.0% [99.5%-100.0%, $I^2$ = 47.8%]), altered sensorium (67.4% [54.9–78.8%, $I^2$ = 93.3%]), headache (65.0% [51.5–77.6%, $I^2$ = 95.1%]) and neck stiffness 56.6% (29.4–80.4%, $I^2$ = 96.3%). Classical features of scrub typhus were infrequently identified; an eschar was found in only 20.8% (9.8%-34.3%, $I^2$ = 95.4%) and lymphadenopathy in 24.1% (95% CI 11.8% - 38.9%, $I^2$ = 87.8%). The pooled CFR (95% CI) was 3.6% (1.5%– 6.4%, $I^2$ = 67.3%). Paediatric cohorts had a CFR of 6.1% (1.9–12.1%, $I^2$ = 77%) whilst adult cohorts reported 2.6% (0.7–5.3%, $I^2$ = 43%).

**Funding:** BDM is supported by the UKRI/MRC (MR/V03605X/1), the MRC/UKRI (MR/V007181//1), MRC (MR/T028750/1) and Wellcome (ISSF201902/3). The funders did not play any role in the study design, data collection and analysis, decision to publish or preparation of the manuscript.

**Competing interests:** The authors have declared that no competing interests exist.

## Conclusion

Our meta-analyses illustrate that 3.6% of patients with CNS manifestations of scrub typhus die. Clinicians should have a high index of suspicion for scrub typhus in patients presenting with AES in endemic regions and consider starting empiric treatment whilst awaiting results of investigations, even in the absence of classical signs such as an eschar or lymphadenopathy.

### Author summary

Scrub typhus is a leading cause of acute encephalitis syndrome in endemic regions, but its clinical features and case fatality rate are not yet fully described. We conducted a systemic review and meta-analysis and found that adults with neurological scrub typhus commonly present with headache and neck stiffness, whilst children present with altered sensorium and seizure activity. Signs classically associated with scrub typhus such as eschar and lymphadenopathy were not commonly reported. Just under 4% of patients with neurological manifestations of scrub typhus die in hospital, and the case fatality rate may be higher in children. Clinicians should have a low threshold to start empiric treatment for scrub typhus when patients present with neurological infections in endemic regions.

## Introduction

Scrub typhus is a zoonotic rickettsial illness caused by the bacteria *Orientia tsutsugamushi*. It is transmitted through larvae (chiggers) of *Leptotrombidium* mites and is endemic to a region termed the *tsutsugamushi triangle*, which spans from south-eastern Asia to the pacific [1]. Chigger bites may result in the pathognomonic eschar forming on the skin [2], and other early symptoms of scrub typhus can range from an acute febrile illness to fulminant multi-organ failure [3].

The central nervous system (CNS) can often be affected in scrub typhus, with neurological manifestations being present in roughly 20% of cases, either in the form of acute encephalitis, meningitis, or as meningoencephalitis [4]. An estimated one million cases of scrub typhus occur annually in the *tsutsugamushi triangle* [5]. As a result, recent epidemiological studies have suggested that scrub typhus is now a leading cause of CNS infections in endemic regions [6–8]. As changing ecology may increase the prevalence of arthropod-borne CNS infections populations globally [9], neurological manifestations of scrub typhus may become an emerging public health beyond current endemic regions.

Globally, few studies have reported the case fatality rate of scrub typhus infection, and even fewer describe the proportions of patients with specific neurological symptoms of the disease. A systematic review on untreated scrub typhus infection suggests a mortality of 6%[5]. However, given that the administration of easily accessible antibiotics leads to considerably improved outcomes, overall mortality rates have the potential to be considerably lower [4]. Delayed treatment can, on the other hand, contribute to significant neurological sequela or even death [10]. Prompt antibiotic therapy relies on early recognition of scrub typhus. Though small, single-site studies from a select few geographical regions have reported the clinical features CNS scrub typhus [8,11–29], these have not been systematically analysed to help understand when clinicians should consider a diagnosis of scrub typhus CNS disease.

We aimed to evaluate the clinical features and case fatality ratio (CFR) of scrub typhus with neurological manifestations through a meta-analysis of published studies. Our objective was to elucidate clinical features that should increase suspicion and therefore prompt empiric treatment of scrub typhus in patients presenting with syndromes of CNS infection.

## Methods

### Search strategy and selection criteria

We conducted a systematic review and meta-analysis of studies investigating neurological manifestations of scrub typhus. We undertook a literature search of PubMed (MEDLINE) and Scopus for English language peer-reviewed primary research articles published between 1st January 2000 and 22nd April 2022. We chose these dates as they both reflect modern scrub typhus management and ease of access to serological tests [30]. Key search terms included scrub typhus, *Orientia tsutsugamushi*, central nervous system, meningitis, encephalitis, and meningoencephalitis (the full search strategy is supplied in S1 Table).

We included studies reporting the hospital mortality of patients with neurological manifestations of scrub typhus. Patients of any age were included if they had either meningitis, encephalitis, encephalopathy, or meningoencephalitis. We excluded studies which were case reports, case series, review articles, conference abstracts and letters. Studies with case numbers below 10 and those not reporting demographic features of their cohorts were excluded to reduce sampling bias (S2 Table).

Titles and abstracts from each database search were imported into EndNote X8 (Clarivate Analytics, Philadelphia, USA) and duplicates were removed. The remaining studies were uploaded to Rayyan (Qatar Computing Research Institute, Doha, Qatar) for screening. Screening of titles and abstracts and subsequent full-texts manuscripts were undertaken in parallel by two authors (AMA and CSG) and conflicts were resolved through consensus. Data were then extracted by the two authors (AMA and CSG). We collected study cohort characteristics and outcomes (a full list of collected data are supplied in S3 Table). Authors of screened articles were contacted to provide additional information when otherwise absent from the publication.

### Diagnosis

Scrub typhus infection was defined as cases confirmed by indirect serology (which included IgM enzyme-linked immunosorbent assay [ELISA], immunofluorescence assay [IFA], immunochromatographic test [ICT] and the Weil–Felix test) or direct diagnosis through polymerase chain reaction [PCR]. Patients with co-infections were excluded. Whilst IFA is considered the gold standard for serological diagnosis, IgM ELISA is more commonly used as it requires no sophisticated instrument facilities, whilst still being a robust diagnostic tool in scrub typhus [31]. In comparison, the Weil Felix test have been reported to show lower specificity and should only be considered significant at higher serum titres [32]. Despite this, the Weil-Felix test is still used in smaller laboratories for screening of scrub typhus in rural areas and can be a useful diagnostic tool in the correct context [33,34]. We conducted sensitivity analysis to explore if the meta-analyses were sensitive to the exclusion of papers using the Weil-Felix test for diagnosis of scrub typhus.

### Outcomes

The primary aim of our meta-analyses was to study the prevalence of the clinical features seen in patients with neurological manifestation of scrub typhus. Secondly, we estimate a weighted

CFR by dividing the number of deaths by the number of patients with proven CNS scrub typhus infections.

## Quality assessment

Quality assessment of individual studies was completed by two authors (AMA and CSG) using a point scale based on the CASP tool for cohort studies [35]. We scored studies on five main aspects: study design and methodology (0–2 points), quality of adjustment for confounding factors (0–2 points), definition of outcome (0–2 points), and quality of data (0–3 points). A maximum of 10 points represented the lowest risk of bias and high methodological quality, whilst a score over 6 was deemed to have low risk of bias, a score of 4–6 as moderate and less than 4 as high risk (S4 Table).

This study followed the guidelines published by the Preferred Reporting Items for Systematic Reviews and Meta-Analyses (PRISMA)[36]. We registered our study protocol prospectively on PROSPERO (ID 328732).

## Statistical analysis

All statistical analyses including meta-analysis was completed using R version 4.0 (R Foundation for Statistical Computing, Vienna, Austria). We used the DerSimonian-Laird random-effects model to report prevalence of clinical variables and weighted pooled CFR. A Freeman-Tukey double arcsine transformation was applied to our data. We conducted sub-group analyses in paediatric (defined as studies solely recruiting patients aged <16 years or from paediatric hospitals) and adult cohorts (defined as studies recruiting patients from general hospitals). Forest plots were generated with natural scales, and heterogeneity was assessed using Cochran's Q test and $I^2$, whilst Egger's test was used to test for publication bias.

## Results

Our search strategy identified 714 records, of which 206 were duplicates. We screened 508 abstracts for inclusion, and of these 148 proceeded to full-text screening. We excluded 129 studies, resulting in 19 eligible studies for meta-analyses (Fig 1).

## Included studies and risk of bias assessment

The 19 included studies were published from 2013 to 2020, reflecting 1,221 patients. The studies were from 3 countries, with 15 from India of 1,124 (92.1%) patients, 1 from Laos of 31 (2.5%) patients, 2 from South Korea of 38 (3.1%) patients and 1 from an unspecified country of 28 (2.3%) patients. Most studies recruited patients in tertiary hospitals (12/19, 63.1%), with 4 (21.1%) recruiting their cohorts from teaching hospitals. There was 1 (5.3%) multicentre study, and 2 (10.5%) studies recruited patients from unspecified units. Of the 19 studies, 11 (57.8%) studies were prospective cohorts, 7 (36.8%) were retrospective cohorts and 2 (10.5%) were cross-sectional studies. Over half of the studies (10/19, 52.9%) included patients with a diagnosis of meningoencephalitis, 6 (31.6%) included those with a diagnosis of meningitis and 3 (15.8%) included patients with encephalitis. Only 12 (63.2%) studies specified the treatment provided to their cohorts. Most studies (12/19, 63.2%) were conducted on an adult cohort whilst the remainder (7/19, 36.8%) specifically recruited in paediatric settings. The mean [SD] study enrolment period was 34.8 [27.8] months.

When conducting quality assessment of the studies, 3 studies had an overall high risk of bias, 11 had moderate risk and 5 had low risk (Fig 2 and S3 Table).

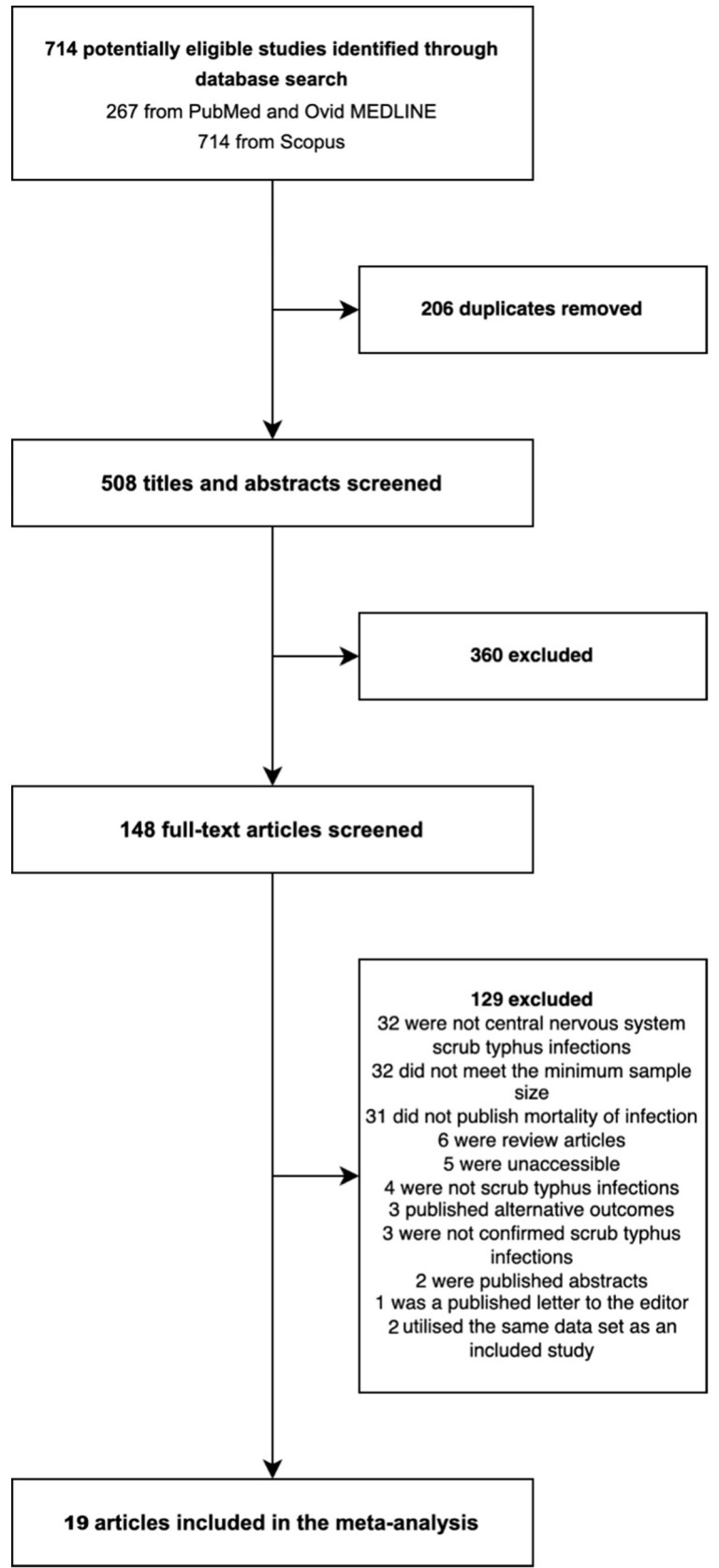

**Fig 1. Study selection.**

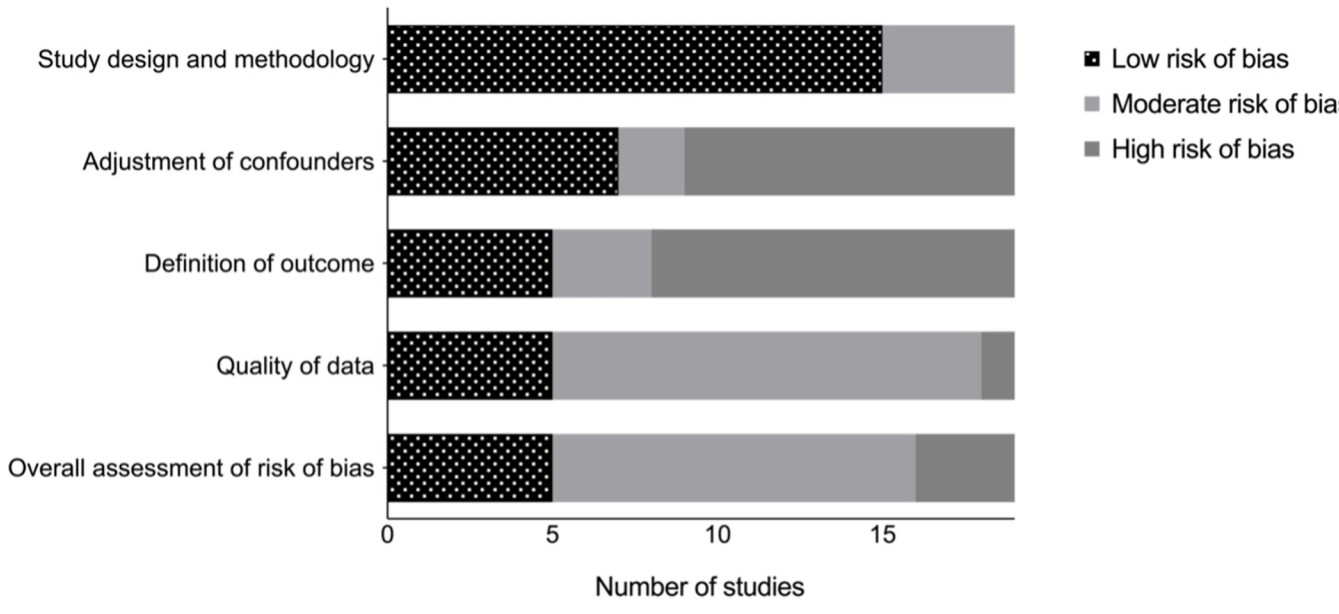

**Fig 2. Quality assessment for the risk of bias in the 19 included studies for meta-analyses.**

## Demographics of included cases

Overall, 1,221 patients with neurological manifestations of scrub typhus were included in the analysis (Table 1). Of these, 656 (53.7%) were adults and 565 (46.3%) were paediatric patients. The mean (SD) age was 29.1 (19.0) and 573 (46.9%) were female. Patients presented to hospital after a median (IQR) duration of symptoms of 7.9 (7.0–9.2) days and had a median (IQR) Glasgow coma scale (GCS) score of 11/15 (9–15). Complete treatment data were available for 859 cases of whom 389 (45.3%) were treated with doxycycline monotherapy and 379 (44.1%) with azithromycin only. Combination therapy of more than one antibiotic was given to 82 (9.5%) patients. A handful received chloramphenicol or rifampicin (four and five respectively).

## Clinical features of neurological manifestations of scrub typhus

The most common clinical features reported were fevers (weighted pooled prevalence [WPP] 100.0%, [95% CI 99.5–100.0%], $I^2$ = 47.8%), altered sensorium (WPP 67.4% [54.9–77.8%], $I^2$ = 93.3%), headache (WPP 65.0%, [51.5–77.6%], $I^2$ = 95.1%) and neck stiffness (WPP 55.6%, [29.4–80.4%], $I^2$ = 96.3%) (Table 2). We found a WPP of seizures of 43.8% ([25.1–63.4%], $I^2$ = 96.8%). Identification of an eschar was found in only 20.8% ([9.8–34.3%], $I^2$ = 95.4%) of the pooled patients. Lymphadenopathy was present in only 24.1% ([11.8–38.9%], $I^2$ = 87.8%). Dyspnoea and cough were present in approximately a quarter of patients (25.2% [10.3–43.6%], $I^2$ = 94.1% and 26.2% [12.6–42.3%], $I^2$ = 88.7% respectively).

In sub-group analysis, the most common symptoms identified in paediatric cohorts of 565 patients were fever in 565 (100.0%), seizures in 417 (83.1%), altered sensorium in 323 (74.1%) and organomegaly in 297 (52.6%). In adult cohorts of 656 patients, the common features reported were fever in 648 (98.8%). Headaches were reported in 424 of the 656 patients (64.6%). Neck stiffness was assessed in 142 patients and reported in 93 (65.5%), and altered sensorium was assessed in 639 and reported in 325 (50.9%).

**Table 1. General characteristics of included studies for meta-analyses;** * = specification of treatment strategy required reporting of what antibiotics (if any) were given to enrolled patients; PCR = polymerase chain reaction; ELISA = enzyme-linked immunosorbent assay, IFA = immunofluorescence assay, ICT = immunochromatographic test, PCR = polymerase chain reaction, WFT = Weil–Felix test.

| Study | Country | Study design | Study setting | Study cohort population | Study enrolment period | Number of patients | Median or mean age (SD) [IQR] | Number of females (%) | Number of deaths (%) | Diagnostic test | Specified treatment strategy* |
|---|---|---|---|---|---|---|---|---|---|---|---|
| Kim 2013 [20] | South Korea | Retrospective cohort | Tertiary hospital | Mixed | 2004–2008 | 22 | 70.2 (Not specified) | 14 (62.6%) | 0 (0.0%) | PCR | Yes |
| Viswanathan 2013 [29] | India | Retrospective cohort | Unspecified | Mixed | 2011–2012 | 17 | 41.8 (17.7) | 7 (41.2%) | 0 (0.0%) | IgM ELISA or WFT | Yes |
| Misra 2014 [21] | India | Cross sectional | Tertiary hospital | Mixed | 2012–2013 | 37 | 37.7 [3 – 71] | 19 (51.4%) | 0 (0.0%) | ICT or WFT | Yes |
| Dittrich 2015 [17] | Laos | Prospective study | Tertiary hospital | Mixed | 2003–2011 | 31 | 16.0 [8.0–30.0] | 9 (29.0%) | 3 (1.0%) | IFA | No |
| Abhilash 2015 [11] | India | Retrospective cohort | Tertiary hospital | Mixed | 2005–2011 | 189 | 41.0 (6.3) | 81 (42.9%) | 11 (5.8%) | IgM ELISA | Yes |
| Sharma 2015 [26] | India | Prospective study | Tertiary hospital | Mixed | 2009–2011 | 23 | 39.5 [Not specified] | 10 (43.5%) | 0 (0.0%) | WFT | Yes |
| Jamil 2015 [19] | India | Prospective study | Tertiary hospital | Mixed | 2013–2014 | 13 | 34.8 (16.2) | 5 (38.5%) | 2 (6.5%) | ICT or WFT | Yes |
| Bhat 2016 [14] | India | Prospective study | Teaching hospital | Paediatric | Unspecified | 27 | 9.4 (5.2) | 11 (40.7%) | 2 (7.4%) | IgM ELISA | No |
| Rana 2016 [24] | India | Prospective study | Teaching hospital | Mixed | 2013–2014 | 37 | 48.1 [22 – 80] | 30 (81.1%) | 0 (0.0%) | IgM ELISA | Yes |
| Lee 2017 [8] | South Korea | Retrospective cohort | Tertiary hospital | Mixed | 2009–2014 | 16 | 35.5 [Not specified] | 10 (62.5%) | 0 (0.0%) | IFA | Yes |
| Rose 2017 [25] | India | Retrospective cohort | Tertiary hospital | Paediatric | 2010–2015 | 63 | 8.9 (4.1) | 39 (61.9%) | 0 (0.0%) | IgM ELISA or WFT | Yes |
| Gangwar 2020 [23] | India | Retrospective cohort | Tertiary hospital | Paediatric | 2018 | 146 | 7.0 [4.0–11.0] | 68 (46.6%) | 13 (8.9%) | IgM ELISA | Yes |
| Thakur 2020 [27] | India | Cross sectional | Multicentre | Mixed | 2013–2018 | 210 | 29.8 (16.6) | 90 (42.9%) | 14 (6.7%) | IgM ELISA | No |
| Valappil 2017 [28] | Unspecified | Prospective study | Unspecified | Mixed | Unspecified | 28 | 40.2 (17.6) | 14 (50.0%) | 0 (0.0%) | IgM ELISA | No |
| Bhat 2017 [15] | India | Prospective study | Teaching hospital | Paediatric | Unspecified | 19 | 7.8 (4.9) | 5 (26.3%) | 1 (5.2%) | IgM ELISA | No |
| Mittal 2018 [22] | India | Retrospective cohort | Tertiary hospital | Paediatric | 2016 | 230 | 5.0 [3.0–10.0] | 116 (50.4%) | 35 (15.2%) | IgM ELISA | Yes |
| Dinesh Kumar 2018 [16] | India | Prospective study | Tertiary hospital | Paediatric | Unspecified | 14 | 6.9 (2.7) | 7 (50.0%) | 0 (0.0%) | IgM ELISA | No |
| Alam 2020 [12] | India | Prospective study | Teaching hospital | Paediatric | 2016–2018 | 66 | 5.0 (3.8) | 20 (30.3%) | 7 (10.6%) | IgM ELISA | Yes |
| Arora 2021 [13] | India | Prospective study | Tertiary hospital | Mixed | 2014–2016 | 33 | 46.5 (17.2) | 18 (54.5%) | 2 ((6.1%) | IgM ELISA | No |

Taken together, adult patients were more likely than children to present with headache, neck stiffness and an eschar. However, even in adult cohorts, an eschar was only identified in one in five cases. Children were more likely to present with altered sensorium, seizures, and organomegaly.

In hospital mortality data were available for 1,221 patients and the number of patients who died with CNS complications of scrub typhus infection was 90 (7.4%). The calculated pooled CFR (95% CI) was 3.6% (1.5–6.4%, $I^2$ = 67.4%). The pooled CFR (95% CI) was 2.4% (0.6–4.9%, $I^2$ = 39.9%) in adult cohorts and higher in the paediatric cohorts with a CFR of 6.1% (1.9–12.1%, $I^2$ = 76.6%) (Fig 3, funnel plots are show in S5 Table).

**Table 2. Weighted pooled prevalence (WPP) of clinical features in patients with neurological manifestations of scrub typhus;** * = Defined as general changes in brain function including abnormal Glasgow coma scale score; Ɨ = defined as hepatomegaly, splenomegaly or both Case fatality ratio.

| | Adult cohorts | Paediatric cohorts | Weighted pooled prevalence | 95% CI | p Value for Cochran's Q | I² | Eggers's test (p value) |
|---|---|---|---|---|---|---|---|
| Fever (%) | 648/656 (98.8%) | 565/565 (100.0%) | 100.0% | 99.5% - 100.0% | 0.012 | 47.8% | 0.004 |
| Headache (%) | 424/656 (64.6%) | 155/565 (27.4%) | 65.0% | 51.5% - 77.6% | <0.001 | 95.1% | 0.004 |
| Altered sensorium (%) * | 325/639 (50.9%) | 323/436 (74.1%) | 67.4% | 54.9% - 78.8% | <0.001 | 93.3% | 0.344 |
| Neck stiffness (%) | 93/142 (65.5%) | 71/337 (21.1%) | 55.6% | 29.4% - 80.4% | <0.001 | 96.3% | 0.001 |
| Seizures (%) | 95/407 (23.3%) | 417/502 (83.1%) | 43.8% | 25.1% - 63.4% | <0.001 | 96.8% | 0.115 |
| Dyspnoea (%) | 148/355 (41.7%) | 44/175 (25.1%) | 25.2% | 10.3% - 43.6% | <0.001 | 94.1% | 0.188 |
| Cough (%) | 72/286 (25.2%) | 23/123 (18.7%) | 26.2% | 12.6% - 42.3% | <0.001 | 88.7% | 0.503 |
| Diarrhoea (%) | 21/247 (8.5%) | 28/439 (6.4%) | 5.8% | 3.4% - 8.6% | 0.182 | 35.9% | 0.598 |
| Abdominal pain (%) | 66/301 (21.9%) | 101/499 (20.2%) | 24.3% | 16.5% - 33.0% | <0.001 | 82.2% | 0.124 |
| Vomiting (%) | 234/543 (43.1%) | 187/392 (47.7%) | 46.2% | 34.1% - 58.5% | <0.001 | 91.6% | 0.740 |
| Jaundice (%) | 9/60 (15.0%) | 23/376 (6.1%) | 8.3% | 2.9% - 15.5% | 0.164 | 72.9% | 0.164 |
| Organomegaly (%) Ɨ | 111/324 (34.3%) | 297/565 (52.6%) | 48.0% | 38.3% - 57.7% | <0.001 | 85.4% | 0.556 |
| Lymphadenopathy (%) | 59/307 (19.2%) | 26/112 (23.3%) | 24.1% | 11.8% - 38.9% | <0.001 | 87.8% | 0.257 |
| Eschar (%) | 147/559 (26.3%) | 31/546 (5.7%) | 20.8% | 9.8% - 34.3% | <0.001 | 95.4% | 0.067 |
| Rash (%) | 97/353 (27.5%) | 50/532 (9.4%) | 20.2% | 11.1% - 31.1% | <0.001 | 91.3% | 0.149 |

## Sensitivity analysis

When conducting sensitivity analysis with and without studies utilising the Weil-Felix test, we found no significant difference in both the weighted pooled prevalence of the clinical features of scrub typhus, and the reported case fatality ratio (S5 Table).

## Discussion

Scrub typhus is an increasingly prevalent cause of neurological infection in endemic regions [9]. In this systematic review and meta-analyses of 20 studies representing 1,221 patients, we investigated the clinical features and CFR in patients with CNS scrub typhus. Most studies had a moderate to high risk of bias, and there was significant heterogeneity between the included cohorts.

We identified that patients with neurological manifestations of scrub typhus present with non-specific AES features, including fever, headache, altered sensorium, neck stiffness and seizures [38]. The 'characteristic' symptoms of scrub typhus were uncommonly reported, with the pathognomonic eschar seen in only 20% of patients and lymphadenopathy being present in 24%. Nevertheless, the eschar may be more common in adults and conversely organomegaly in children. However, as these features are reported in only one in four to five patients with proven CNS scrub typhus, the absence of these signs should not be used to rule out scrub typhus infection. In comparison, approximately 25% of patients reported non-specific respiratory symptoms such as shortness of breath and cough–a finding described in other aetiologies of CNS infections such as *Haemophilus influenzae*, cryptococcus and Nipah virus [39–41].

In paediatric cohorts, organomegaly was common and may signify a useful clinical marker. Organomegaly in scrub typhus is not well reported in literature but has often been described

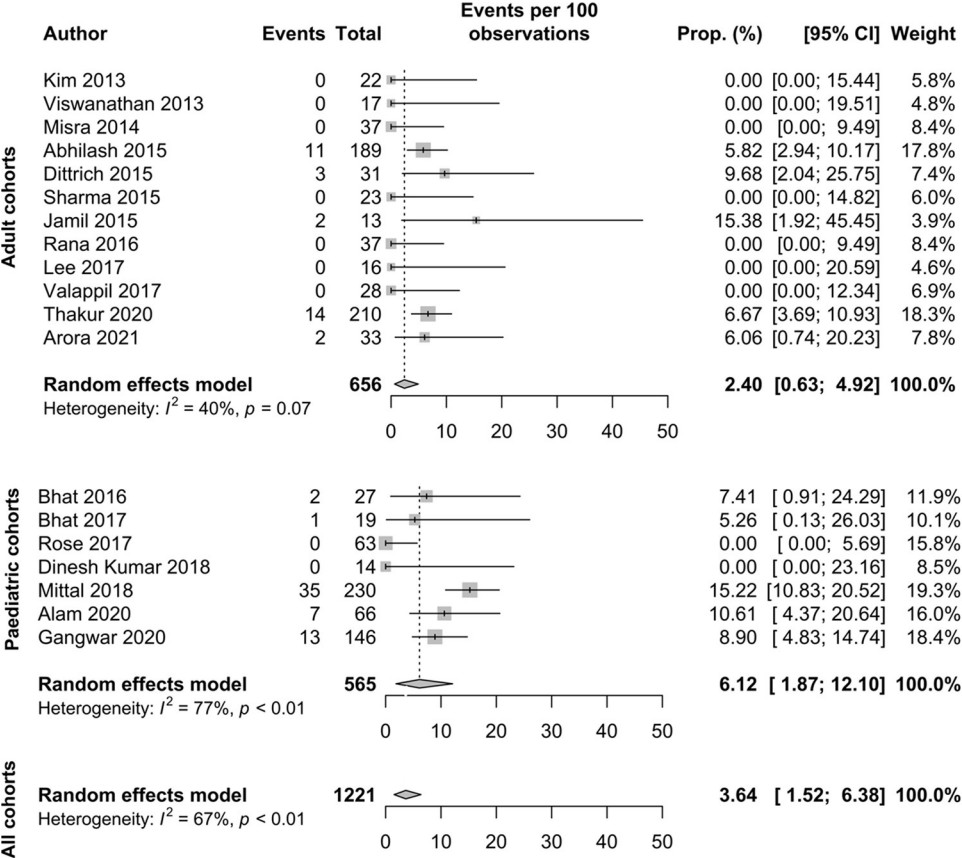

**Fig 3. Forest plot showing case fatality ratio for adult, paediatric and combined cohorts; events are defined as number of in hospital fatalities [8,11–17,19–22,24–26,28,29,37].**

in other systemic bacterial infections such as typhoid and tuberculosis [42]; hepatomegaly specifically has been reported to be present in up to one in ten patients with CNS infections [43]. Presence of organomegaly may suggest systemic and disseminated scrub typhus in children, and therefore act as an indicator that the CNS may also be affected. This is of particular importance given some children within our meta-analysis did not present with signs suggestive of AES (such as headaches or altered sensorium), but may have substantial neurological involvement given the majority reported seizures. Seizure activity can be common in AES in children [44] and may be in part due to the increased occurrence of febrile convulsions, rather than neurological inflammation itself [45]. Nevertheless, clinicians should be alert to the potential for acute seizure activity in children with CNS scrub typhus given its potential effect on long-term morbidity [46].

Our meta-analyses suggests that approximately 3.6 per 100 patients with scrub typhus neurological infection die in hospital. This CFR was comparable to a study investigating outcomes of all-cause CNS infections among 725 patients in a tropical region [47]. However, our pooled CFR was not as high as those reported for other aetiologies of CNS infections such as herpes simplex virus (HSV) encephalitis [48,49], meningococcal meningitis [48] and Japanese encephalitis [50]. In the studies included in our analyses, most patients were treated with appropriate antibiotics (such as doxycycline, azithromycin, rifampicin, or chloramphenicol) and our CFR was roughly half of that seen in untreated scrub typhus infection [5]. Tetracyclines, and doxycycline in particular, have good CNS penetration [51] and can lead to an early

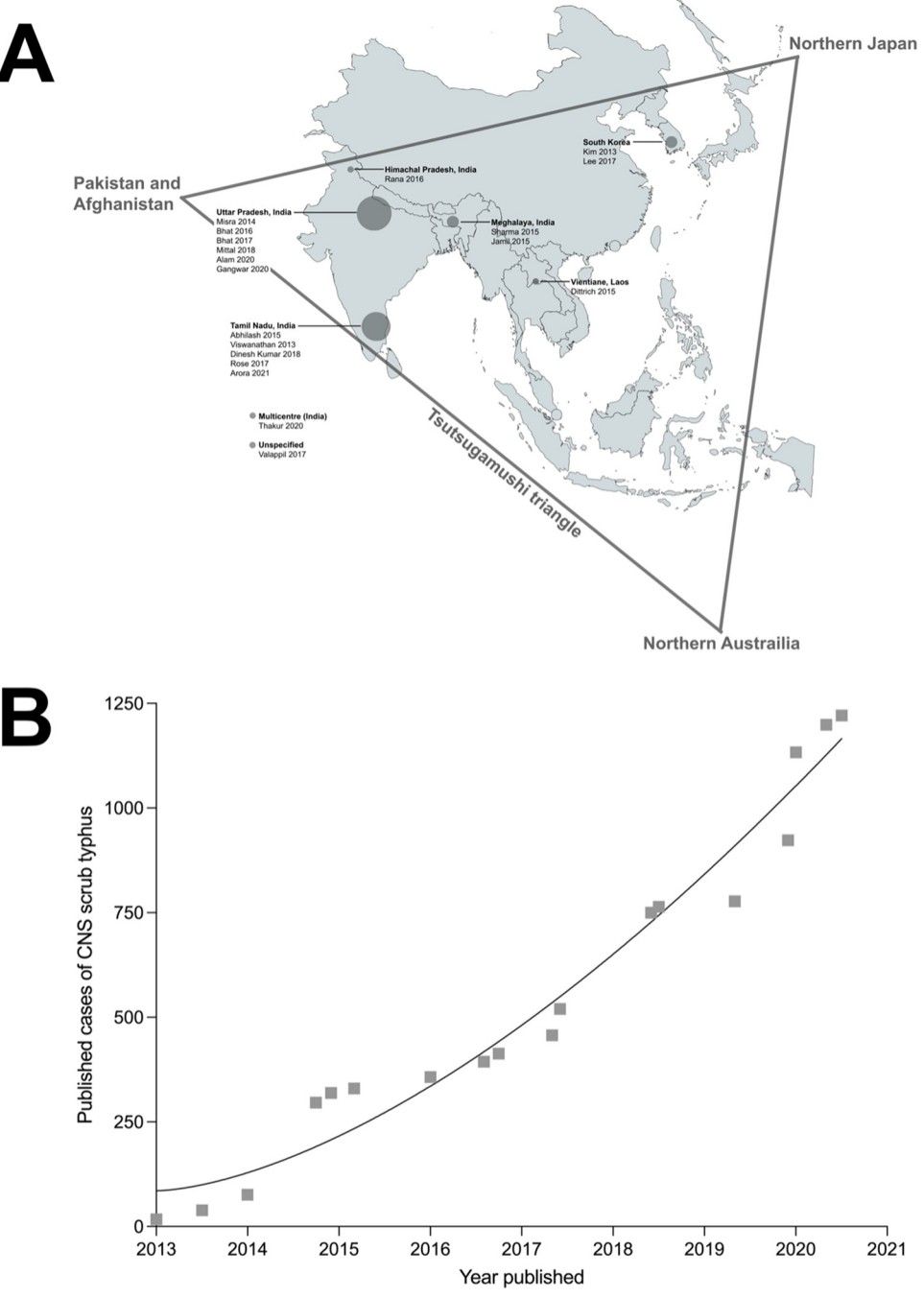

**Fig 4.** A: Map illustrating *tsustugamushi triangle* and the included studies in our meta-analyses, circles are proportionate to the size of the cohorts. The map was created using public domain image (https://commons.wikimedia.org/wiki/File:BlankMap-World.svg). B: Graph illustrating number of published cases of central nervous system (CNS) scrub typhus described in literature.

cure of severe scrub typhus with neurological involvement [52]. Early antibiotic therapy remains integral to ensuring improved outcomes in scrub typhus [53].

We observed that CFR may be higher in paediatric cohorts when compared with adult cohorts. The CFR in paediatric cohorts was 6.1%, which corroborates studies examining

outcomes in all-cause CNS infections in paediatric cohorts [54,55]. We considered treatment difference as a cause for the differing CFR–paediatric cohorts were often treated with azithromycin rather than doxycycline, likely due to concerns regarding the side-effects of tetracyclines in children [56]. However, a meta-analysis has illustrated that there is no observable difference between antibiotics for treatment of scrub typhus, with all options producing favourable outcomes [53]. Therefore, an alternate explanation we suggest is that there may be a greater delay in diagnosis in paediatric cohorts. The clinical presentations of all-cause CNS infections in young children can be nonspecific; features suggestive of AES were not common among paediatric patients. A high index of suspicion and syndromic approach is essential as delay in initiation of antibiotics is associated with increased mortality in bacterial infections of the CNS in children [57,58]. Given the propensity of scrub typhus to present with diverse clinical features, identifying behavioural and geographical risk factors to indicate the possibility of chigger bites is important in diagnosis [59]. In children, this may be harder to obtain, further delaying the diagnosis. Furthermore, paediatric cohorts in our study reported higher proportions of seizures which could contribute to mortality, whilst the prevalence of organomegaly in paediatric cohorts could suggest that higher proportions of disseminated scrub typhus infections occur in children [60].

## Limitations and future direction

Our study had several limitations. Our analyses were based on estimates derived from cohort studies which often did not report potential confounders for fatality. For example, co-morbidities (especially those causing immunocompromise) have been shown to be associated with worse outcomes in patients with neurological infections [61]. Furthermore, very few studies specified a timescale for their cohorts, and both these factors may bias results for our calculated fatality. We explored heterogeneity using predefined analyses and though the between-study heterogeneity we report may represent true variation in clinical features and CFR, difference in methodology and study quality may confound this measure. We included serological tests given their prevalence in endemic regions, but acknowledge that they are imperfect and their accuracy is highly dependent on the cut-offs used for diagnosis. Finally, though we explored the clinical features present in patients with scrub typhus CNS infections, we did not investigate blood studies, cerebrospinal fluid or imaging findings in these patients. Some studies in our meta-analyses reported these findings, and there is scope to analyse investigation findings in scrub typhus to elucidate characteristics features when compared to other CNS infections.

Several factors may contribute to the possibility of our pooled CFR being an underestimate of the true CFR in CNS scrub typhus. Our studies were centred around six sites, with many studies recruiting patients from the same sites (Fig 4A). Furthermore, all studies included in our meta-analyses recruited from either tertiary or teaching hospitals and the clinicians at sites recruiting patients may be more adept at recognising and managing neurological scrub typhus infections. This reduces the generalisability of our results and fatality may be higher in settings where fewer specialists are available; more studies are needed in other settings in the *tsusugamushi triangle*, particularly from rural regions where burden of scrub typhus is high [12,62].

Increasing numbers of CNS scrub typhus have been described in the literature (Fig 4B). However, only one included study in our analyses reported the long term sequalae of neurological scrub typhus infection [23]. With greater access to acyclovir, there has been drastic improvement in mortality in patients with HSV encephalitis. However, with this improved mortality, more patients have been found to suffer from prolonged neurological deficits [63]. Similarly, given scrub typhus's excellent response to antibiotics, there is the possibility that many patients suffer from long-term neurological morbidity. Further research is required to

quantify the burden of morbidity. Additionally, there is little known about the specific cause of death in those with neurological manifestations of scrub typhus. Some studies report that meningoencephalitis is a predictor of poorer outcome in patients [64], but few detail why this may be. With seizures being common in our analyses, the possible implication of subsequent neurological complications such as status epilepticus on fatality should be investigated. Identifying and intervening in treatable causes of death may have substantial effects on the long-term prognosis and survival of this population.

## Conclusion

In this study of neurological manifestations of scrub typhus, we found an overall CFR of 3.8%, which may be highest in children. Adults were more likely to present with headache and neck stiffness, whereas children were more likely to present with altered sensorium and seizures. Nevertheless, clinical features were often non-specific; the pathognomonic eschar was only present in a minority, especially in children, and should not be used to rule out scrub typhus infection. In endemic areas clinicians should have a low threshold for empirical antibiotic therapy in patients presenting with AES, whilst undertaking investigations for alternative aetiologies. Further high-quality evidence which focuses on long-term morbidity in patients with neurological manifestations of scrub typhus are required.

## Supporting information

**S1 Table. Search strategy.**
(DOCX)

**S2 Table. Inclusion criteria.**
(DOCX)

**S3 Table. Data collected.**
(DOCX)

**S4 Table. Bias assessment.**
(DOCX)

**S5 Table. Sensitivity analysis (excluding studies which used Weil-Felix test in diagnosis).**
(DOCX)

**S1 Fig. Funnel plots.**
(DOCX)

## Author Contributions

**Conceptualization:** Ali M. Alam, Jack Goodall.

**Data curation:** Ali M. Alam.

**Formal analysis:** Ali M. Alam, Conor S. Gillespie.

**Investigation:** Benedict D. Michael.

**Methodology:** Conor S. Gillespie, Benedict D. Michael.

**Supervision:** Jack Goodall, Tina Damodar, Lance Turtle, Ravi Vasanthapuram, Tom Solomon, Benedict D. Michael.

**Validation:** Conor S. Gillespie, Jack Goodall.

**Writing – original draft:** Ali M. Alam, Jack Goodall, Tina Damodar, Benedict D. Michael.

**Writing – review & editing:** Ali M. Alam, Conor S. Gillespie, Jack Goodall, Tina Damodar, Lance Turtle, Ravi Vasanthapuram, Tom Solomon, Benedict D. Michael.

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
