## [Decision Letter · Decision Letter 0]

14 Oct 2022

Dear Dr Alam,

Thank you very much for submitting your manuscript "Neurological manifestations of scrub typhus infection: a systematic review and meta-analysis of clinical features and case fatality" for consideration at PLOS Neglected Tropical Diseases. As with all papers reviewed by the journal, your manuscript was reviewed by members of the editorial board and by several independent reviewers. In light of the reviews (below this email), we would like to invite the resubmission of a significantly-revised version that takes into account the reviewers' comments. 

We cannot make any decision about publication until we have seen the revised manuscript and your response to the reviewers' comments. Your revised manuscript is also likely to be sent to reviewers for further evaluation.

Sincerely,

Joseph Raymond Zunt

Academic Editor

Joseph Vinetz

Section Editor

Reviewer's Responses to Questions

**Key Review Criteria Required for Acceptance?**

**Methods**

-Are the objectives of the study clearly articulated with a clear testable hypothesis stated?

-Is the study design appropriate to address the stated objectives?

-Is the population clearly described and appropriate for the hypothesis being tested?

-Is the sample size sufficient to ensure adequate power to address the hypothesis being tested?

-Were correct statistical analysis used to support conclusions?

-Are there concerns about ethical or regulatory requirements being met?

Reviewer #1: -Are the objectives of the study clearly articulated with a clear testable hypothesis stated? Yes

-Is the study design appropriate to address the stated objectives? No

-Is the population clearly described and appropriate for the hypothesis being tested? No (some queries need to be addressed)

-Is the sample size sufficient to ensure adequate power to address the hypothesis being tested? Yes

-Were correct statistical analysis used to support conclusions? Yes

-Are there concerns about ethical or regulatory requirements being met? Yes

Reviewer #2: This is a well-planned meta-analysis with clear aims stated. The authors ensured the inclusion of the most appropriate studies using a robust criteria. The analyses were described and suitable for the aims stated.

**Results**

-Does the analysis presented match the analysis plan?

-Are the results clearly and completely presented?

-Are the figures (Tables, Images) of sufficient quality for clarity?

Reviewer #1: -Does the analysis presented match the analysis plan? Yes

-Are the results clearly and completely presented? Yes

-Are the figures (Tables, Images) of sufficient quality for clarity? Yes

Reviewer #2: The results are presented in a clear manner and supplemented by the figures included.

**Conclusions**

-Are the conclusions supported by the data presented?

-Are the limitations of analysis clearly described?

-Do the authors discuss how these data can be helpful to advance our understanding of the topic under study?

-Is public health relevance addressed?

Reviewer #1: -Are the conclusions supported by the data presented? No

-Are the limitations of analysis clearly described? No

-Do the authors discuss how these data can be helpful to advance our understanding of the topic under study? Yes

-Is public health relevance addressed? Yes

Reviewer #2: The conclusions match the results found with clear limitations conveyed.

**Editorial and Data Presentation Modifications?**

Reviewer #1: (No Response)

Reviewer #2: Nil

**Summary and General Comments**

Reviewer #1: The authors have conducted a systematic review and meta analysis to report the clinical features and case fatality ratio in scrub typhus patients. They have conducted the review according to their hypothesis. I have a few queries:

1. Is lymphadenopathy a pathognomonic feature of scrub typhus? Please give references for this.

2. In para 4 of Introduction, authors mention inclusion of case series in their review but in para 2 of methods, they have mentioned excluding case series. This needs clarification.

3. Methodology: Scrub typhus diagnosed by IFA, Weil-Felix and ELISA have been included in these study for review. The specificity of Weil-Felix is poor and should ideally not be included in review such as these. The authors should take a look at their data after removing patients diagnosed solely by Weil-Felix test.

4. Results: Dyspnoea was present in 25.1% of patients. Was this ARDS? If yes, then it would be difficult to delineate in which patients mortality was due to CNS disease versus ARDS. The authors have not mentioned anything related to this. 

5. Results: In Table 1 and figure 3, the studies mentioned as Kumar 2018, Jamil 2019 and Arora 2020 are missing from the reference list.

Reviewer #2: Scrub typhus is an important but neglected infectious disease and this meta-analysis on the neurological manifestations of scrub typhus is extremely welcomed. The authors have included studies based on a robust criteria and although the included studies exhibited heterogeneity and risk of bias, some conclusions could be drawn from the analysed data. I have some comments to add:

- The serological assays used (please correct in the text - indirect immunofluorescence assay, Weil-Felix test and IgM ELISA) are imperfect and accuracy highly dependent on the cut-offs used for diagnosis (if based on an acute sample only rather than paired acute and convalescent samples). The Weil-Felix test is no longer widely utilised due to lack of sensitivity and specificity. In India, where most of the included studies were from, the InBios Scrub Typhus IgM ELISA is widely used and there have been debate around an appropriate OD cut-off if based on a single acute sample. 

It would be helpful to allude to the imperfections of serological testing in the manuscript. Also, did some studies include molecular assays or culture? If so, was there concordance with the serological results?

- Historically, sensori-neural hearing loss was described in patients with scrub typhus. Were there any descriptions of this in the included studies?

- In the included studies, was testing of CSF samples performed? If so, it could be clinically helpful to include the cell count pattern, protein and glucose levels in scrub typhus patients with CNS involvement, even if molecular testing for Orientia tsutsugamushi is unavailable.

- With regards to treatment, it is important to clarify route of antibiotic administration as drug absorption through the gastrointestinal route may be impaired in critically unwell patients with organ dysfunction. There is also a discrepancy in the availability of parenteral antibiotics between regions - e.g. IV doxycycline is widely available in India but often not elsewhere. Did the included studies clarify this?

PLOS authors have the option to publish the peer review history of their article (what does this mean?). If published, this will include your full peer review and any attached files.

Reviewer #1: No

Reviewer #2: No
---

## [Editor Report · Decision Letter 1]

7 Nov 2022

Dear Dr Alam,

We are pleased to inform you that your manuscript 'Neurological manifestations of scrub typhus infection: a systematic review and meta-analysis of clinical features and case fatality' has been provisionally accepted for publication in PLOS Neglected Tropical Diseases.

Best regards,

Joseph Raymond Zunt

Academic Editor

Joseph Vinetz

Section Editor

---

## [Editor Report · Acceptance letter]

24 Nov 2022

Dear Dr Alam,

We are delighted to inform you that your manuscript, "Neurological manifestations of scrub typhus infection: a systematic review and meta-analysis of clinical features and case fatality," has been formally accepted for publication in PLOS Neglected Tropical Diseases.

Best regards,

Shaden Kamhawi

co-Editor-in-Chief

Paul Brindley

co-Editor-in-Chief
